# Organomegalies as a predictive indicator of leukemia cutis in patients with acute myeloid leukemia

**Chanakarn Kanitthamniyom**[1‡], **Chalothorn Wannaphut**[2‡], **Penvadee Pattanaprichakul**[3], **Smith Kungwankiattichi**[1,4], **Weerapat Owattanapanich**[1,4]*

**1** Division of Hematology, Department of Medicine, Faculty of Medicine Siriraj Hospital, Mahidol University, Bangkok, Thailand, **2** Department of Medicine, John A. Burns School of Medicine, University of Hawaii, Honolulu, Hawaii, United States of America, **3** Department of Dermatology, Faculty of Medicine Siriraj Hospital, Mahidol University, Bangkok, Thailand, **4** Center of Excellence of Siriraj Adult Acute Myeloid/Lymphoblastic Leukemia (SiAML), Faculty of Medicine Siriraj Hospital, Mahidol University, Bangkok, Thailand

‡ CK and CW contributed equally to the manuscript as first authors.
* weerapato36733@gmail.com

## Abstract

### Background

Leukemia cutis (LC) is an extramedullary acute myeloid leukemia (AML) infiltrate. No previous study has described the clinical characteristics and outcomes of Thai patients diagnosed with AML with LC.

### Materials and methods

We conducted a 7-year retrospective case-control study on Thai AML patients at Siriraj Hospital from November 2013 to July 2020. Patients were divided into LC and non-LC groups. Initial clinical presentations and laboratory findings were examined to identify LC-associated factors. Overall survival (OS) and relapse-free survival (RFS) were assessed. Pathological tissues underwent re-evaluation to validate the LC diagnoses.

### Results

The study included 159 patients in a 2:1 ratio (106 non-LC and 53 LC). The LC group had a mean ± SD age of 54.3 ± 15.5 years; females were predominant. Three-fifths of the LC patients had intermediate-risk cytogenetics; 20.4% had an adverse risk, and 10.2% had a favorable risk. Most were classified as AML-M4 and AML-M5. Leukemic nodules were the primary finding in 58.5% of the cases, mainly on the legs. In the multivariate analysis of predictive factors associated with LC, organomegalies, specifically hepatomegaly, and lymphadenopathy, remained significant factors associated with LC [OR 4.45 (95%CI 1.20, 16.50); p = 0.026 and OR 5.48 (95%CI 1.65, 18.20); p = 0.005], respectively. The LC group demonstrated a significantly reduced OS (log-rank test p = 0.002) (median OS of 8.6 months vs. 32.4 months). RFS was considerably lower in the LC group (log-rank test p = 0.001) (median duration of 10.3 months vs. 24.4 months in the non-LC).

**Data Availability Statement:** All relevant data are within the paper and its Supporting Information files.

**Funding:** The author(s) received no specific funding for this work.

**Competing interests:** The authors have declared that no competing interests exist.

## Conclusions

AML patients who developed LC tended to experience notably poorer prognoses. Therefore, it is imperative to consider aggressive treatment options for such individuals. The presence of organomegalies in AML patients serves as a strong predictor of the possible occurrence of LC when accompanied by skin lesions.

## Introduction

Acute myeloid leukemia (AML) is a hematologic malignancy originating within the bone marrow. While patients primarily present with bone marrow failure symptoms [1, 2], extramedullary manifestations, such as myeloid sarcoma or leukemia cutis (LC), are also observed [3–7]. These signs often signify a poor prognosis and reduced survival rates [8–22].

LC is characterized by the extramedullary infiltration of myeloid leukemia cells into the epidermis, dermis, and subcutaneous tissue [23–27]. Notably, AML is chiefly associated with LC. Approximately 50% of the AML cases presenting with LC are classified as subtype M4 (acute myelomonocytic leukemia) and subtype M5 (acute monocytic leukemia) under the French-American-British (FAB) classification system [12, 13]. The significant extramedullary involvement in M4/M5 AML is attributed to the larger size and shorter half-life of monocyte white blood cells [12–17]. These cutaneous manifestations are often associated with poor overall survival (OS) outcomes [8–14]. Frequently, LC co-exists with extramedullary infiltrations in other organs, such as the central nervous system, liver, gums, and lymph nodes [28–31]. These extramedullary phenomena might be detected at the initial diagnosis or upon disease recurrence [32–34].

Prior research has uncovered a spectrum of diverse cutaneous manifestations related to LC, including trisomy 8 and *MLL* gene rearrangement (11q23) as common genetic markers in AML patients with LC [35–43]. Despite these insights, a comprehensive analysis of the clinical characteristics of LC in Thai AML patients has never been reported, and research on molecular mutations within this patient population remains scant. Accordingly, this study compared the OS between AML groups, specifically distinguishing those with skin involvement from those without such involvement. Moreover, the study set out to identify significant determinants, such as clinical characteristics, laboratory results, chromosomal aberrations, and especially molecular abnormalities, to differentiate the LC-afflicted groups from non-LC groups.

## Materials and methods

A seven-year retrospective study was conducted on Thai patients diagnosed with AML at Siriraj Hospital, Bangkok, Thailand, between November 2013 and July 2020. Siriraj Hospital is a tertiary referral center. The inclusion criteria comprised patients aged 15 or above diagnosed with AML according to the 2016 World Health Organization classification and treated at the Hematology Clinic of Siriraj Hospital. Acute promyelocytic leukemia diagnoses were excluded. Eligible patients were identified using ICD-10 codes (C92.4 to C92.9, C93.0, C94.0, C94.2, and C95.0). A matched case-control study involving 159 patients was conducted. The case group was the AML patients with LC, while the control group was the AML patients without LC. Cases and controls were matched on baseline age, sex, the 2017 European LeukemiaNet (ELN) classification [2], and AML type, which is *de novo* or secondary. The primary objective was to differentiate the OS outcome of patients with AML who had LC from those who did not present with LC. The secondary objective was to identify predictive factors that could distinguish between these two groups of patients. A two-block randomization method was used to stratify patients in a 2:1 ratio between these categories. Detailed data were gathered, including baseline

characteristics (i.e., age, sex, and comorbidities), clinical characteristics of AML, treatment approaches, outcomes, cytogenetic information, and specific skin findings in LC patients. The Ethics Committee approved the study protocol for Research in Human Subjects at the Faculty of Medicine, Siriraj Hospital, Mahidol University (approval code Si 935/2020). The study was performed in accordance with the Declaration of Helsinki. The data were accessed after obtaining ethical approval for research on November 30, 2020. Data collection was conducted without disclosing any information that could identify individual patients, and the data were encrypted as electronic files accessible only to the research team.

## Definitions

Hepatomegaly was defined as a median liver span exceeding 15 cm as measured using transabdominal ultrasound in the midclavicular line [44]. Splenomegaly was determined when the spleen's vertical length exceeded 13 cm [45]. Lymphadenopathy was characterized by lymph node enlargement defined as a diameter exceeding 1.5 cm [45].

## Statistical analysis

The baseline variables, including sex, age, and pre-existing conditions, were characterized through descriptive statistics. Categorical data are reported as frequency (%), continuous data are reported as mean ± SD for normally distributed data, and median (interquartile range) for non-normally distributed data. The Kolmogorov-Smirnov test examined the normality of distribution of variables. Determinants influencing survival were assessed by the Chi-squared test or Fisher's exact test for categorical data as appropriate and by the independent t-test for continuous data. The primary endpoint—OS in both AML groups—was tested for equality of OS survival function between groups by the log-rank test. For only patients receiving intensive chemotherapy, a propensity score-adjusted Kaplan–Meier survival curve by inverse probability of treatment weighted (IPTW) Kaplan–Meier estimation adjusted for age at diagnosis, sex, and cytogenetic risk by the 2017 ELN classification [2] was performed with stabilized weights and hypothesis testing for significant difference in curves of outcomes performed by bootstrap test at 5000 replicates for OS and relapse-free survival (RFS), comparing the outcomes between the AML with LC group and AML without LC group. A 99% trimmed weights sensitivity analysis of IPTW Kaplan–Meier estimation was performed [46]. We performed Cox proportional hazard for the effect size of LC on OS, including propensity score matching, adjusting the same factors as the propensity score-adjusted Kaplan-Meier estimates at a 1:3 ratio without replacement by greedy matching with a caliper width 0.2 x SD of the logit of the PS [47]. A standardized mean difference <0.1 was considered an acceptable balance on covariates. Acceptable covariate balance for PS matching for RFS could be obtained by optimal matching with exact matching on cytogenetic risk stratum at a 1:2 ratio without replacement. We further compared the AML group without LC to the AML group with LC using logistic regression to identify factors associated with LC. All statistical analyses were executed with PASW Statistics (version 18; SPSS Inc, Chicago, IL), Stata 14.0 (StataCorp. College Station, TX), and propensity score analyses by R 4.2.0 (R Core Team 2023, R Foundation for Statistical Computing, Vienna, Austria) using the adjustedCurves [48], matchit, and survival packages.

## Results

### Baseline characteristics of AML patients with LC

The 53 AML patients with LC had a mean ± SD age of 54.3 ± 15.5 years, and 52.8% were female. *De novo* AML occurred in 73.6% (39 of 53 patients), and the remaining 14 cases were

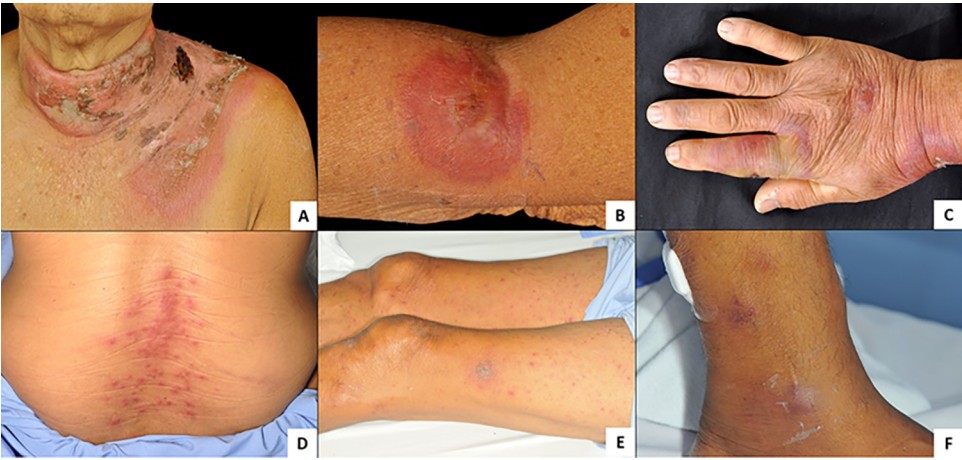

**Fig 1. Cutaneous manifestations of acute myeloid leukemia patients with leukemia cutis.** (A) Cellulitis-like plaque with desquamation on the neck and shoulder, (B) an indurated plaque with a vesicular center on the arm, (C) nodules and plaques on the dorsum of the hand, (D) multiple erythematous papules on lower back, bilateral multiple discrete erythematous papules and plaques on both legs, (F) discrete erythematous nodules around the ankle.

secondary AML, including 9 MDS-related cases, 2 therapy-related cases, and 3 cases from other categories. Within the LC group, the FAB classifications predominantly placed patients in the AML M4 and M5 subtypes (60.3%). According to the 2017 ELN classification, risk stratification showed that 69.4% (34 patients) had intermediate risk, 20.4% had an adverse risk, and 10.2% had favorable risk. According to the Eastern Cooperative Oncology Group performance status (ECOG-PS), the distribution was ECOG-PS 0 (1.9%), ECOG-PS 1 (64.2%), ECOG-PS 2 (17%), ECOG-PS 3 (13.2%), and ECOG-PS 4 (3.8%).

Dermatologic manifestations were diverse, with skin nodules being the most common (58.5%), followed by plaques (13.2%), papules (11.3%), and ulcers (9.4%). These predominantly occurred on the upper (42%) and lower extremities (50.9%). **Figs 1** and **2** illustrate the dermatologic and dermatopathologic findings. Splenomegaly, lymphadenopathy, and hepatomegaly were noted in 12, 10, and 9 patients. **Table 1** summarizes the baseline characteristics of the dermatologic lesions.

The common cytogenetic findings were normal karyotype (57.8%), complex cytogenetics (11.1%), t(16::16) translocation (6.7%), +8 trisomy (4.4%), t(8::21) translocation (2.2%), and 11q23 rearrangement (2.2%). Molecular analysis using targeted next-generation sequencing was performed on 9 of the 53 patients. On average, 2 mutations were identified per patient. The most frequent mutations were *NRAS* (15%), *DNMT3A* (10%), *NPM1* (10%), *JAK2* (10%), and *SRSF2* (10%) (**Fig 3**).

## Comparison of baseline characteristics and outcomes in patients with and without leukemia cutis

The study enrolled 159 AML patients and divided them into two groups: the AML with LC group (53 patients) and the AML without LC group (106 patients). The allocation followed a 2:1 ratio with careful matching of age, sex, and AML subtype between the two groups. The patient flow through the study is depicted as a flow chart in **Fig 4**.

Both groups exhibited slight female predominance (54.7%). A significant difference between the LC and non-LC groups was observed in the prevalence of hepatomegaly, splenomegaly, and lymphadenopathy, with p-values of 0.011, 0.008, and 0.005, respectively. A

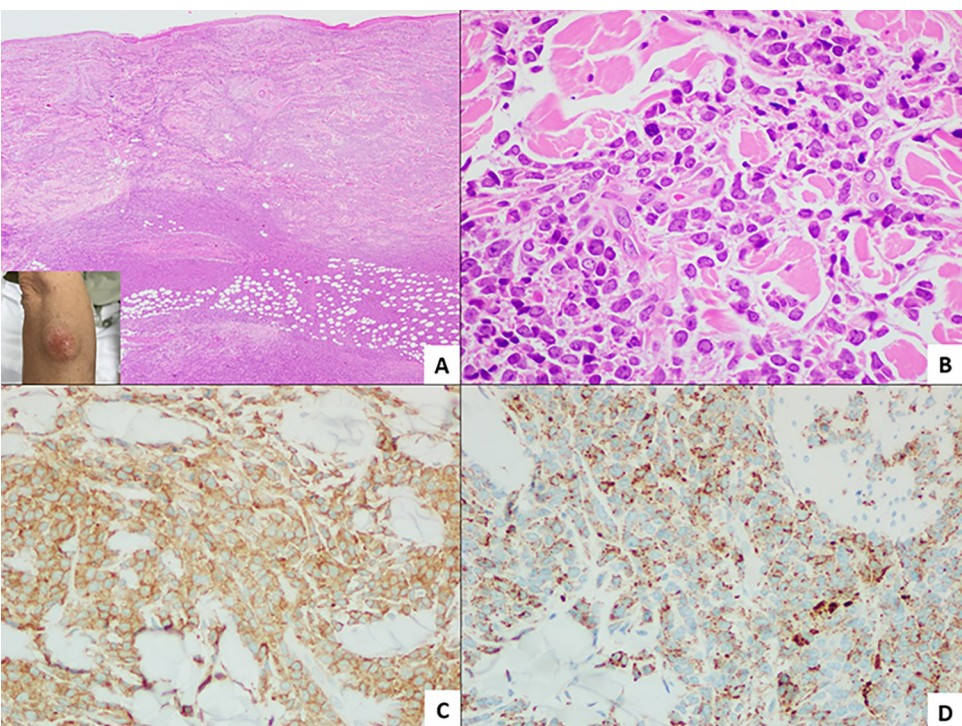

**Fig 2. Histopathologic findings of AML (M4) in a patient with leukemia cutis presenting with a large nodule on the forearm (inset A).** A. Diffuse dermal to subcutaneous infiltrates by atypical myeloid cells with grenz zone. (hematoxylin & eosin, x 20 low power field magnification); B. Infiltrates comprised medium to large-sized mononuclear cells with blastoid features. (hematoxylin & eosin, x 600 high-power field magnification); C and D. Myeloid infiltrates were reactive for CD33 (C, x 400 high power field magnification) and lysozyme. (D, x 400 high power field magnification).

significant discriminative difference in the FAB classification of AML was demonstrated, with M4/M5 being more prevalent in LC patients (p = 0.001). There was a significant difference in median peripheral blood blast percentage between the LC and non-LC groups. The LC group's median peripheral blood blast percentage (interquartile range) was measured at 15% (0, 49),

**Table 1. Prevalence of skin lesions in 53 acute myeloid leukemia patients with leukemia cutis.**

| Characteristics | n (%) |
| --- | --- |
| Skin lesion | |
| Nodule | 31 (58.5) |
| Plaque | 7 (13.2) |
| Papule | 6 (11.3) |
| Ulcer | 5 (9.4) |
| Patch | 4 (7.5) |
| Macule | 1 (1.9) |
| Location of the skin lesion | |
| Head and neck | 11 (20.8) |
| Trunk | 20 (37.7) |
| Arms | 21 (39.6) |
| Legs | 27 (50.9) |
| Palms | 1 (1.9) |

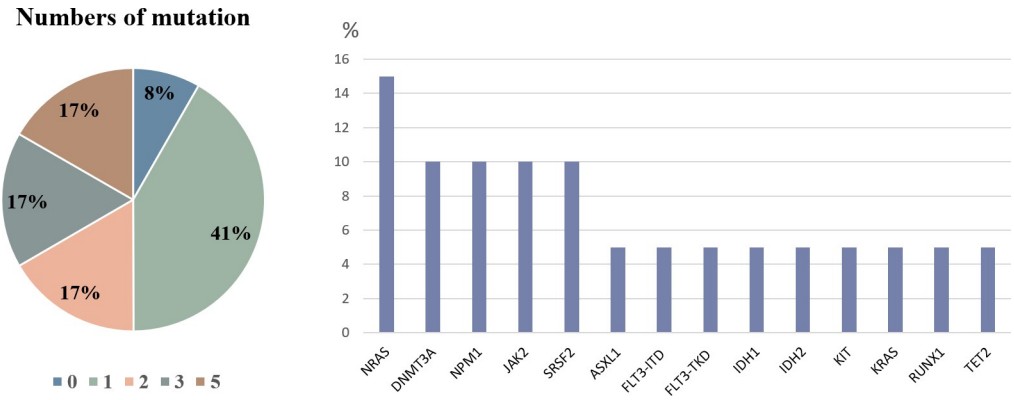

**Fig 3. Prevalence of genetic mutations in leukemia cutis patients.**

and the non-LC group's percentage (interquartile range) was 41% (14, 78) (p < 0.001). However, the median bone marrow blast percentages did not differ significantly between groups.

The AML patients received a variety of treatment approaches. These included intensive chemotherapy protocols like the 3+7 regimen (59.0%) and the 2+5 regimen (4.5%). The acute promyelocytic leukemia (APL) protocol, which utilizes the ATRA-idarubicin combination based on the AIDA 0493 protocol [49], was employed in 1.9% of cases. Other treatments were transfusion support (19.9%), hypomethylating agents (8.3%), hydroxyurea (3.8%), and low-dose cytarabine (2.6%). In the pooled group treated with low-dose cytarabine, hypomethylating agents, and intensive chemotherapy protocols, including the APL protocol, the non-LC group did not have a significant difference in complete remission rate compared to the LC group [OR 1.12 (95%CI 0.47, 2.66); p = 0.802]. Results for complete remission rates for the intensive chemotherapy group and hypomethylating agents/low-dose cytarabine group separately were also not significantly different between the non-LC group and the LC group [OR 1.03 (95%CI 0.40, 2.68); p = 0.950, and OR 1.00 (95%CI 0.08, 12.6); p = 1.00, respectively].

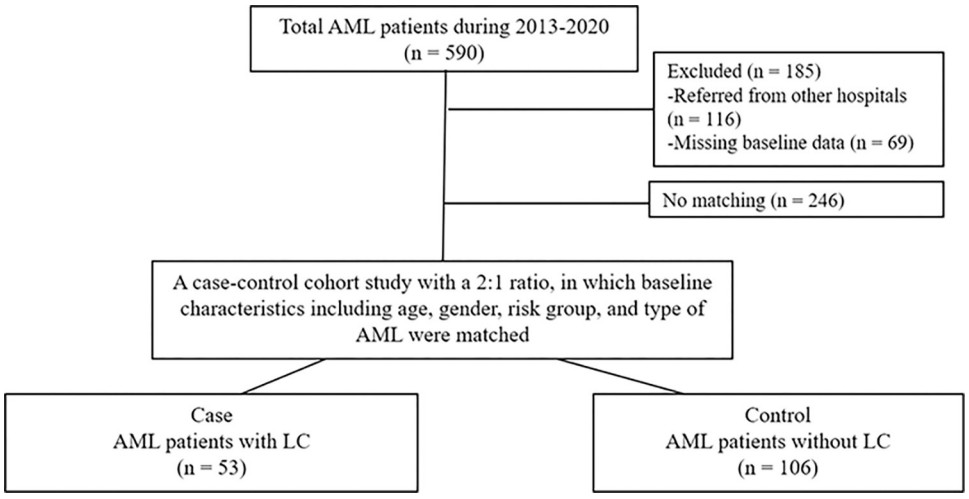

**Fig 4. Flow chart of patients in the study.**

**Table 2. Comparative analysis of clinical characteristics and laboratory findings in leukemia cutis and non-leukemia cutis acute myeloid leukemia patients.**

| Characteristics | Total (N = 159) | Leukemia cutis group (n = 53) | Non-leukemia cutis group (n = 106) | P-value |
|---|---|---|---|---|
| Age, mean ± SD, y | 53.5 ± 15.4 | 54.3 ± 15.5 | 53.1 ± 15.5 | 0.625 |
| Sex | | | | 0.735 |
| Male | 72 (45.3) | 25 (47.2) | 47 (47.2) | – |
| Female | 87 (54.7) | 28 (52.8) | 59 (52.9) | – |
| Type of AML | | | | 0.599 |
| *De novo* | 121 (76.1) | 39 (73.6) | 82 (77.4) | – |
| Secondary | 38 (23.9) | 14 (26.4) | 24 (22.6) | – |
| FAB type of AML | | | | **0.001** |
| M4 | 50 (31.4) | 20 (37.7) | 30 (28.3) | |
| M5 | 17 (10.7) | 12 (22.6) | 5 (4.7) | |
| Non M4–5 | 92 (57.9) | 21 (39.6) | 71 (67.0) | |
| Hepatomegaly | 13 (8.2) | 9 (17.0) | 4 (3.8) | **0.011** |
| Splenomegaly | 20 (12.7) | 12 (22.6) | 8 (7.7) | **0.008** |
| Lymphadenopathy | 15 (9.6) | 10 (18.9) | 5 (4.8) | **0.005** |
| Hb level, mean ± SD, g/dL | 7.9 ± 2.2 | 8.2 ± 2.2 | 7.7 ± 2.2 | 0.175 |
| Median WBC count, median (interquartile range), x $10^9$/L | 16.7 (4.5, 57.6) | 13 (3.3, 51.5) | 20.3 (4.8, 58.0) | 0.326 |
| Peripheral blood blasts, median (interquartile range), % | 30 (7, 68) | 15 (0, 49) | 41 (14, 78) | **< 0.001** |
| Median platelet count, median (interquartile range), x $10^9$/L | 50 (27, 93) | 54 (29, 116) | 46 (26, 81) | 0.139 |
| Median bone marrow blasts, median (interquartile range), % | 70 (40, 90) | 70 (23, 80) | 70 (40, 90) | 0.082 |
| Fibrinogen level, mean ± SD (mg/dL) | 382.7 ± 163.3 | 385.6 ± 158.4 | 381.4 ± 166.7 | 0.903 |
| D-dimer level, median (interquartile range), ng/mL | 2689.5 (948.9, 7136.1) | 3156.0 (1096, 4772) | 2256.4 (942.8, 8385) | 0.916 |

**Notes:-** Data are presented as n (%) unless stated otherwise. P-values significant at the <0.05 level are denoted in bold font.

**Abbreviations:** *AML* acute myeloid leukemia, *FAB* French-American-British classification system, *Hb* hemoglobin, *WBC* white blood cell count.

**Table 2** provides a comparative overview of the clinical characteristics and laboratory findings comparing the LC and non-LC patients.

Additionally, we conducted a mutational profiling comparison between both groups. The prevalence of almost all mutations was similar between the two groups, but the *NRAS* mutation was significantly more prevalent in the LC group (33.3% vs. 9.0%; p = 0.023). Treatment regimens and the proportion of patients undergoing allogeneic stem cell transplants were similar between both groups (**Table 3**).

The LC group demonstrated a significantly reduced median OS of 8.6 months compared to 32.4 months in the non-LC group (log-rank test p = 0.002) [hazard ratio (HR) 1.98 (95%CI 1.28, 3.09); p = 0.002] (**Fig 5A**). RFS was also notably significantly lower in the LC group with a median duration of 10.3 months versus 24.4 months in the non-LC group (log-rank test p = 0.001) [HR 2.40 (95%CI 1.37, 4.18), p = 0.002] (**Fig 5B**).

In our multivariate logistic regression analysis of predictive factors for LC, we identified significant associations with AML subtypes M4/M5 and peripheral blood blast counts of less than 20%. Additionally, after adjustment for the FAB subtype of AML and peripheral blood blast level, we found that organomegalies of hepatomegaly and lymphadenopathy remained significant predictive factors for LC [OR 4.45 (95% CI 1.20, 16.5), p = 0.026 and OR 5.48 (95% CI 1.65, 18.2, p = 0.005), respectively]. Moreover, splenomegaly showed a trend to significance with LC [OR 2.72 (95% CI 0.98, 7.54); p = 0.054]. Detailed results of the univariate and multivariate analyses of factors associated with LC can be found in **Table 4**.

**Table 3. Comparative analysis of risk stratification, genetic profiling, and treatments in leukemia cutis and non-leukemia cutis acute myeloid leukemia patients.**

| | Leukemia cutis group | Non-leukemia cutis group | P-value |
|---|---|---|---|
| The 2017 ELN risk stratification | | | 0.988 |
| Favorable | 5/49 (10.2) | 10/103 (9.7) | |
| Intermediate | 34/49 (69.4) | 71/103 (68.9) | |
| Adverse | 10/49 (20.4) | 22/103 (21.4) | |
| Mutations | | | |
| *ASXL1* | 1/9 (11.1) | 6/89 (6.7) | 0.624 |
| *CBL* | 0/9 (0) | 3/89 (3.4) | 0.864 |
| *CEBPA* | 0/9 (0) | 12/89 (13.5) | 0.466 |
| *DNMT3A* | 2/9 (22.2) | 18/89 (20.2) | 0.886 |
| *EZH2* | 0/9 (0) | 6/89 (6.7) | 0.797 |
| *FLT3*-ITD | 1/11 (9.1) | 23/90 (25.6) | 0.286 |
| *FLT3*-TKD | 1/9 (11.1) | 10/89 (11.2) | 0.991 |
| *IDH1* | 1/9 (11.1) | 4/89 (4.5) | 0.394 |
| *IDH2* | 1/9 (11.1) | 6/89 (6.7) | 0.624 |
| *KIT* | 1/9 (11.1) | 11/89 (12.4) | 0.913 |
| *KRAS* | 1/9 (11.1) | 9/89 (10.1) | 0.924 |
| *TET2* | 1/9 (11.1) | 19/89 (21.3) | 0.498 |
| *RUNX1* | 1/9 (11.1) | 15/89 (16.9) | 0.668 |
| *NPM1* | 2/12 (16.7) | 17/90 (18.9) | 0.854 |
| *NRAS* | 3/9 (33.3) | 8/89 (9.0) | **0.023** |
| *U2AF1* | 0/9 (0) | 6/89 (6.7) | 0.797 |
| *SRSF2* | 2/9 (22.2) | 5/89 (5.6) | 0.070 |
| *ZRSR2* | 0/9 (0) | 1/89 (1.1) | 0.491 |
| *TP53* | 0/9 (0) | 9/89 (10.1) | 0.596 |
| Treatment | | | |
| Hypomethylating agent or low dose cytarabine | 5/53 (9.4) | 12/103 (11.7) | 0.674 |
| Intensive chemotherapy (3+7) regimen, 2+5 regimen, or APL induction protocol) | 33/53 (62.3) | 69/103 (67.0) | 0.557 |
| Allogeneic stem cell transplant | 6/53 (11.3) | 11/106 (10.4) | 0.855 |
| Complete remission rate | | | |
| Hypomethylating agent or low-dose cytarabine [a] | 2/5 (40.0) | 2/5 (40.0) | 1.00 |
| Intensive chemotherapy (3+7 regimen, 2+5 regimen or APL induction protocol) | 24/33 (72.7) | 44/60 (73.3) | 0.950 |

**Notes:-** Data are presented as n/N (%) unless stated otherwise. P-values significant at the <0.05 level are denoted in bold font. [a] Limited data due to no bone marrow response evaluation

**Abbreviations:** *APL* acute promyelocytic leukemia, *ELN* European LeukemiaNet.

### Propensity score-adjusted analyses of overall survival and relapse-free survival in the intensive chemotherapy group

For propensity score case-control-matched analyses on OS, there were initially 69 non-LC controls and 31 LC patients for matching, and 17 controls and 5 cases could not be matched. For RFS, there were initially 55 non-LC controls and 24 LC cases, and 9 controls could not be reached while all cases could be matched. All matching variables showed a standardized mean difference of < 0.1, reflecting an acceptable covariate balance (**S1 Table**).

The LC group demonstrated a significantly reduced median OS of 11.2 months (95%CI 7.0, 34.1), while more than half the non-LC group were alive by around 32 months of follow-up.

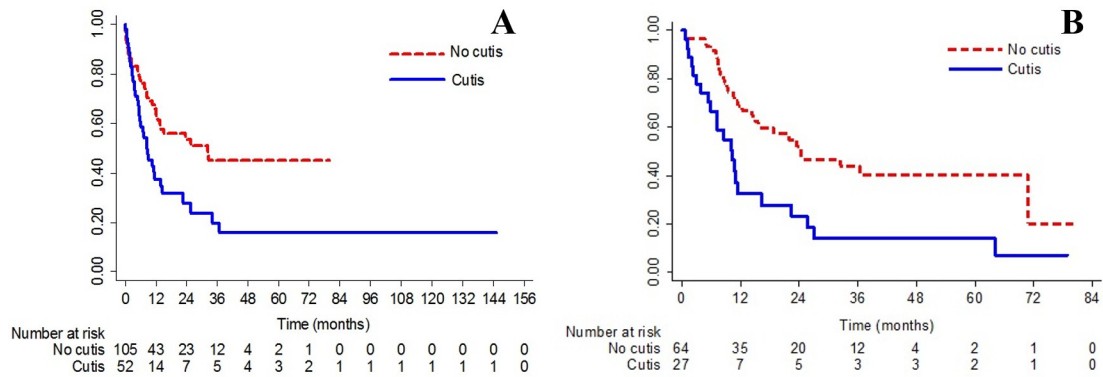

**Fig 5. Kaplan–Meier survival curves illustrate leukemia cutis' survival outcomes compared to non-leukemia cutis AML patients.** (A) Overall survival. (B) Relapse-free survival.

Thus, a median OS was not reached by the end of the period of observation (95%CI 25.6, -) (p = 0.015) (**S1 Fig**) [HR 2.91 (95%CI 1.39, 6.05); p = 0.004]. RFS was also lower in the LC group with a median RFS of 10.2 months (95%CI 5.97, 11.4) versus 36.5 months (95%CI 14.9, -) in the non-LC group (p = 0.012) (**S2 Fig**) [HR 2.76 (95%CI 1.47, 5.18); p = 0.002]. Sensitivity analysis results by 99% trimmed weights for IPTW Kaplan-Meier estimate were similar to the main analysis.

**Table 4. Factors associated with leukemia cutis.**

| Variables | Univariate analysis | |
|---|---|---|
| | **Crude OR (95%CI)** | **P-value** |
| **Hepatomegaly** | **5.16 (1.51–17.7)** | **0.009** |
| Splenomegaly | 3.51 (1.33, 9.23) | **0.011** |
| Lymphadenopathy | 4.65 (1.48, 14.3) | **0.008** |
| FAB M4/M5 | 3.09 (1.56, 6.12) | **0.001** |
| Peripheral blood blasts <20% | 2.84 (1.43, 5.64) | **0.003** |
| **Variables** | **Multivariate analysis** | |
| | **Adjusted OR (95%CI)** | **P-value** |
| **Model 1** | | |
| Hepatomegaly | 4.45 (1.21, 16.5) | **0.026** |
| Peripheral blood blasts <20% | 2.26 (1.09, 4.65) | **0.027** |
| FAB M4/M5 | 2.63 (1.28, 5.41) | **0.008** |
| **Model 2** | | |
| Splenomegaly | 2.72 (0.98, 7.54) | 0.054 |
| Peripheral blood blasts <20% | 2.41 (1.17, 4.96) | **0.017** |
| FAB M4/M5 | 2.57 (1.26, 5.26) | **0.010** |
| **Model 3** | | |
| Lymphadenopathy | 5.48 (1.66, 18.2) | **0.005** |
| Peripheral blood blasts <20% | 2.80 (1.34, 5.86) | **0.006** |
| FAB M4/M5 | 2.77 (1.33, 5.76) | **0.006** |

**Notes:-** P-values significant at the <0.05 level are denoted in bold font.

**Abbreviations:** *FAB* French-American-British classification system

## Discussion

This retrospective case-control study aimed to identify predictive factors in AML patients with LC by comparing them with AML patients without LC. Our results offer concordances and disparities with the literature [12–20, 22]. Our age and sex correlations are consistent with the findings of Agis, et al. and Wang, et al. [12, 13]. Our study also corroborates that FAB-M4 and M5 are the prevailing subtypes in AML patients exhibiting LC. However, we have identified a higher prevalence of LC in AML exhibiting myelomonocytic differentiation (FAB M4), which contrasts with the commonly reported association with monocytic differentiation (FAB M5) in previous studies [12, 13]. Moreover, the pronounced difference in the median peripheral blood blasts between the LC and non-LC groups is a significant observation, diverging from earlier findings that generally indicate higher blast levels in non-LC groups [50]. We hypothesize that the phenomenon of lower blast counts in the peripheral blood of patients with LC compared to non-LC patients can be attributed to several factors. One plausible explanation lies in the preferential sequestration and infiltration of leukemic cells, especially AML M4/M5, within the cutaneous tissue, leading to a reduced presence of these cells in the peripheral blood. Furthermore, the distribution and trafficking of leukemic cells may vary among different tissues. Since the skin is a distinct organ, it may serve as a specific sanctuary site for leukemic infiltration, influencing the distribution pattern of leukemic cells throughout the body. This preferential migration to the skin can contribute to the observed disparity in peripheral blood blast counts between LC and non-LC patients. In addition, our analysis revealed no significant differences in risk stratification based on the 2017 ELN classification between the two groups, similar to the findings of Wang, et al. [13].

When evaluating the manifestations of LC within our AML patients, our findings had some similarities and differences compared to the literature [12–20, 22]. Our study identified nodules as the dominant form of LC skin lesions, principally manifesting on the lower extremities. This observation aligns with the findings of Bakst, et al. [51]. The present study delineates the molecular mutation landscape within a subset of Thai patients with AML manifesting as LC. The investigation found that the *NRAS* mutation was the predominant genetic aberration in AML patients with LC. *NRAS* mutations are known to be prevalent in certain hematologic malignancies, including AML and acute lymphoblastic leukemia. Several studies have suggested that specific genetic mutations, including those involving RAS pathway genes such as *NRAS*, could influence the extramedullary involvement of leukemia, potentially including the skin. The RAS pathway is involved in cell signaling, and mutations in *NRAS* can lead to aberrant activation of this pathway, contributing to the pathogenesis of certain cancers [52–54]. This observation contrasts with Wang, et al. [13], who identified the *NPM1* mutation as the principal genetic marker.

Our research constitutes a case-control study in which we meticulously matched baseline characteristics, including age, gender, and the type of AML, prior to dividing our study into two distinct groups: those with LC and those without LC. This stringent matching process was implemented to mitigate potential confounding factors and additional propensity score analyses adjusting for age, gender, and cytogenetic risk in intensive chemotherapy patients. Furthermore, it is worth noting that our study represents the inaugural investigation into predictive factors amongst those at the initial presentation and the fundamental laboratory findings, specifically focusing on the distinction between patients who present with AML with LC and those who do not. An important observation from our investigation is that enlarged organs, particularly the liver or lymph node, strongly indicate the presence of LC. LC represents an instance of extramedullary involvement in patients with AML, wherein leukemic cells infiltrate the cutaneous tissue. This manifestation occurs in conjunction with the broader

pathophysiological process observed when leukemic cells infiltrate other visceral organs, leading to clinical presentations such as lymphadenopathy, splenomegaly, and hepatomegaly. The connection between organomegaly and LC might be explained by the extramedullary involvement of AML in specific organs [7]. The intricate interplay between systemic disease progression and the infiltration of leukemic cells into the skin underscores the multifaceted nature of acute leukemia's impact on various organ systems. Our study suggests that a skin biopsy should be promptly performed to confirm LC in newly diagnosed AML patients with organomegaly and concomitant skin lesions. Moreover, if an AML patient in complete remission demonstrates new organ enlargement accompanied by the appearance of a skin lesion, this may indicate disease recurrence. Such an observation underscores the critical need for a skin biopsy to corroborate a relapse.

A primary point of contention arises from our study's findings, which demonstrated that AML patients with LC typically significantly reduced both OS and RFS compared to patients without LC. These outcomes align with the results of previous investigations [12, 13]. Notably, a recent study conducted by Dinardo, et al. reported a prevalence of extramedullary blast involvement of 9%, and they demonstrated that aggressive treatment involving venetoclax in combination with FLAG-IDA yielded a near 90% complete response rate [55]. As such, exploring novel therapeutic options for this particular patient subgroup is imperative.

Our study has some limitations. First, its retrospective design may have introduced selection bias and uncontrolled confounders. Second, the retrospective methodology may have led to missing essential information from the medical charts, thereby affecting our conclusions. The heterogeneity of non-randomized treatment regimens across patients may have further influenced the observed outcomes. Additionally, our molecular investigations involving targeted next-generation sequencing were undertaken on a limited subset, encompassing only 11 out of the 53 patients within the LC group. This sample size might not comprehensively represent the molecular characteristics of the entire LC group. To address this limitation, we are conducting an ongoing systematic review and meta-analysis to elucidate the precise incidence of genetic profiling in AML patients presenting with extramedullary blasts compared to those without such manifestations. Lastly, since risk stratification, according to the 2017 ELN classification, primarily relies on genetic mutations and cytogenetic findings, the absence of molecular results may not accurately represent risk group categorization.

## Conclusion

Patients with AML with LC demonstrated a higher prevalence of extramedullary organ involvement and experienced reduced OS and RFS compared to those without LC. Identifying organomegaly as a good predictor of LC in AML may significantly influence clinical decision-making.

## Supporting information

**S1 Table. Balance diagnostics for case-control matching by propensity score.**
(DOCX)

**S1 Fig. Propensity score-adjusted Kaplan–Meier survival curve illustrating overall survival comparing leukemia cutis to non-leukemia cutis AML patients receiving intensive chemotherapy.**
(DOCX)

**S2 Fig. Propensity score-adjusted Kaplan–Meier survival curve illustrating relapse-free survival comparing leukemia cutis to non-leukemia cutis AML patients receiving intensive**

**chemotherapy.**
(DOCX)

## Acknowledgments

The authors express their gratitude to Dr. Anthony Tan, Ms. Kemjira Karnketklang, and Ms. Pataraporn Tunsing for their collaboration in the statistical analyses. Dr. Anthony Tan edited the English language.

## Author Contributions

**Conceptualization:** Chanakarn Kanitthamniyom, Chalothorn Wannaphut, Penvadee Pattanaprichakul, Smith Kungwankiattichi, Weerapat Owattanapanich.

**Data curation:** Chanakarn Kanitthamniyom, Chalothorn Wannaphut, Penvadee Pattanaprichakul, Smith Kungwankiattichi, Weerapat Owattanapanich.

**Formal analysis:** Chalothorn Wannaphut, Weerapat Owattanapanich.

**Investigation:** Chanakarn Kanitthamniyom, Chalothorn Wannaphut, Weerapat Owattanapanich.

**Methodology:** Chanakarn Kanitthamniyom, Weerapat Owattanapanich.

**Project administration:** Weerapat Owattanapanich.

**Resources:** Chanakarn Kanitthamniyom, Penvadee Pattanaprichakul.

**Supervision:** Weerapat Owattanapanich.

**Validation:** Weerapat Owattanapanich.

**Visualization:** Penvadee Pattanaprichakul.

**Writing – original draft:** Chanakarn Kanitthamniyom, Chalothorn Wannaphut, Weerapat Owattanapanich.

**Writing – review & editing:** Chalothorn Wannaphut, Penvadee Pattanaprichakul, Smith Kungwankiattichi, Weerapat Owattanapanich.

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
