## [Decision Letter · Decision Letter 0]

8 Dec 2023

PONE-D-23-37425Organomegalies as a Predictive Indicator of Leukemia Cutis in Patients With Acute Myeloid LeukemiaPLOS ONE

Dear Dr. Owattanapanich,

Thank you for submitting your manuscript to PLOS ONE. After careful consideration, we feel that it has merit but does not fully meet PLOS ONE’s publication criteria as it currently stands. Therefore, we invite you to submit a revised version of the manuscript that addresses the points raised during the review process. Please submit your revised manuscript by Jan 22 2024 11:59PM. If you will need more time than this to complete your revisions, please reply to this message or contact the journal office at plosone@plos.org. Please include the following items when submitting your revised manuscript:A rebuttal letter that responds to each point raised by the academic editor and reviewer(s). You should upload this letter as a separate file labeled 'Response to Reviewers'.A marked-up copy of your manuscript that highlights changes made to the original version. You should upload this as a separate file labeled 'Revised Manuscript with Track Changes'.An unmarked version of your revised paper without tracked changes. You should upload this as a separate file labeled 'Manuscript'.

We look forward to receiving your revised manuscript.

Kind regards,

Mehmet Baysal

Academic Editor

PLOS ONE

Journal Requirements:

Additional Editor Comments:

Experst in the field have evaluated your manuscript. In the light of this reviews; the result is that you should revise the manuscript.

Reviewers' comments:

Reviewer's Responses to Questions

**Comments to the Author**

1. Is the manuscript technically sound, and do the data support the conclusions?

Reviewer #1: Yes

Reviewer #2: Yes

Reviewer #3: Yes

Reviewer #4: Partly

2. Has the statistical analysis been performed appropriately and rigorously? 

Reviewer #1: Yes

Reviewer #2: Yes

Reviewer #3: Yes

Reviewer #4: No

3. Have the authors made all data underlying the findings in their manuscript fully available?

Reviewer #1: Yes

Reviewer #2: Yes

Reviewer #3: Yes

Reviewer #4: Yes

4. Is the manuscript presented in an intelligible fashion and written in standard English?

Reviewer #1: Yes

Reviewer #2: Yes

Reviewer #3: Yes

Reviewer #4: Yes

5. Review Comments to the Author

Reviewer #1: The authors found the relationship with organomegary and cutis.

Is there any biological link between the two?

Cutis may be related to the aggressiveness of leukemia, while organomegary is related to leukemia.

ie. It is described as disease status is related to cutis.

The authors should emphaseze why cutis is related to organomegary

Reviewer #2: The authors describe a large case control study of patients with leukemia cutis. The manuscript is well written and the findings are relevant to the field, supported by the data and are well discused.

Reviewer #3: This is an interesting report indicating organomegaly as a predictor of leukemia cutis (LC) in patients with AML. The study was well organized and nicely written. There are some points that need to be clarified and discussed by the authors:

1. Why patients with LC had lower blast counts in the peripheral blood compared to non-LC patients? Is there any explanation for that?

2. Was there any relationship between NRAS mutation and occurrence of LC? Was it reported in the literature? It needs to be discussed more.

Minor comments:

It is better to present the tables and figures in a separate file.

Reviewer #4: in this work Owattanapanich and cols have analyzed the biological and clinical characteristics of a cohort of AML patients according to the existence of leukemia cutis (LC). To do so, they have performed a case control study obtaining 53 LC-AML patients and 106 non-LC patients. From the clinical point of view, their main findings are the significant association of LC with the presence of adenopathy, splenomegaly, hepatomegaly, low blast peripheral count and monocytic subtype. All of these factors predict the existence of LC. On the other hand, the authors found lower survival and response rate in these subgroup of extramedullary AML.

Major comments

1. What is the main endpoint of this study?

In line 61 the authors state that primary objective is to identify predictive factors discriminating patients with AML with LC from those who did not have LC. However, in the statistical analysis section (line 100) the authors say that the primary endpoint is overall survival in both AML groups. In the case OS is the main endpoint of the study, please change primary objective to in order to identify in lines 76-77. Otherwise, clarify that OS is not the main endpoint of the study.

2. If one of the endpoints of this work was to analyze the predictive value of LC on overall survival, then the method for matching cases and controls is not correct, because it must be adjusted by genetics (which is the main determinant of outcome in AML). Moreover, since NGS was not done in 83% of the LC patients, there could be a bias in such a way that more LC patients harbored worse genetics and this may be the principal reason for their worse outcome. The authors recognize this limitation, but a different analysis could be made to solve this problem. For example, a propensity score matching adjusting for age, sex and genetics according ELN 2016 (or even cytogenetics alone) and performed only in the subgroup of patients treated with intensive chemotherapy could get over this limitation.

3. If NGS was performed in 9/53 LC patients, then the ELN2022 cannot be used for categorizing the genetic risk. Instead, the authors should categorize patients according ELN2016 or simply use cytogenetics.

Minor comments

1. The conclusion in the abstract: “AML patients who developed LC tended to experience notably poorer prognoses” is not sustained by the data presented in the results section of the abstract. The authors should show in the abstract the results corresponding to this conclusion.

2 Line 52 “These cutaneous manifestations are often associated with advanced disease”. The authors might specify the meaning of “advance disease. Is it poor overall survival?, high relapse rate?, poor cytogenetics?.

3.Line 76: Cases and controls were matched on baseline age, sex, and AML type. Please, specify what do you mean by type of AML (further in the text it is obvious that it means de novo vs secondary), but the authors should clarify this criteria the first time this concept is presented in the manuscript.

4. Line 120 “Most patients in the LC group were classified as ECOG 1 (64,2%)” is already written in the previous line. Please remove the repeated sentence.

5. Please show a separate analysis of treatment response in patients treated with intensive chemotherapy and those treated with hypomethylating agents or low dose cytarabine.

6. PLOS authors have the option to publish the peer review history of their article (what does this mean?). If published, this will include your full peer review and any attached files.

Reviewer #1: No

Reviewer #2: **Yes: **Jan Vydra

Reviewer #3: **Yes: **Mohammadreza Bordbar

Reviewer #4: No

---

## [Author Response · Author response to Decision Letter 0]

5 Jan 2024

Plos ONE

January 6, 2024

RE: PLOS ONE-37425 “Organomegalies as a Predictive Indicator of Leukemia Cutis in the

Patients with Acute Myeloid Leukemia”

Dear Editor,

We really appreciate the kind and considerate review of our work and thank the reviewers for the suggested refinements to the manuscript. 

We have endeavored to address the comments. Please see our accompanying point-by-point response, with tracked changes highlighted in yellow in the revised manuscript.

Kind regards,

Weerapat Owattanapanich, MD

Associate Professor of Medicine 

Division of Hematology, Department of Medicine

Faculty of Medicine Siriraj Hospital, Mahidol University

Email: weerapato36733@gmail.com

 

RESPONSE TO REVIEWERS

Reviewer #1

The authors found the relationship with organomegary and cutis.

Comment 1. Is there any biological link between the two? Cutis may be related to the aggressiveness of leukemia, while organomegary is related to leukemia.ie. It is described as disease status is related to cutis. The authors should emphaseze why cutis is related to organomegary.

RESPONSE: Thank you for your insightful suggestion. Leukemia cutis (LC) represents an instance of extramedullary involvement in patients with acute myeloid leukemia (AML), wherein leukemic cells infiltrate the cutaneous tissue. This manifestation occurs in conjunction with the broader pathophysiological process observed when leukemic cells infiltrate other visceral organs, leading to clinical presentations, such as lymphadenopathy, splenomegaly, and hepatomegaly. The connection between organomegaly and LC might be explained by the extramedullary involvement of AML in specific organs. We have added this point in the discussion part.

Manuscript file (Discussion section: page 16, lines 296-301). 

Reviewer #2: 

The authors describe a large case control study of patients with leukemia cutis. The manuscript is well written and the findings are relevant to the field, supported by the data and are well discussed.

RESPONSE: Thank you for your kind comment.

Reviewer #3

This is an interesting report indicating organomegaly as a predictor of leukemia cutis (LC) in patients with AML. The study was well organized and nicely written. There are some points that need to be clarified and discussed by the authors:

Comment 1. Why patients with LC had lower blast counts in the peripheral blood compared to non-LC patients? Is there any explanation for that?

RESPONSE: We hypothesize that the phenomenon of lower blast counts in the peripheral blood of patients with LC compared to non-LC patients can be attributed to several factors. One plausible explanation lies in the preferential sequestration and infiltration of leukemic cells, especially AML M4/M5 within the cutaneous tissue, leading to a reduced presence of these cells in the peripheral blood. Additionally, the distribution and trafficking of leukemic cells may vary among different tissues. Since the skin is a distinct organ, it may serve as a specific sanctuary site for leukemic infiltration, influencing the distribution pattern of leukemic cells throughout the body. This preferential migration to the skin could contribute to the observed disparity in peripheral blood blast counts between LC and non-LC patients. We have added this point in the discussion part.

Manuscript file (Discussion section: pages 16, lines 263-272). 

Comment 2. Was there any relationship between NRAS mutation and occurrence of LC? Was it reported in the literature? It needs to be discussed more.

RESPONSE: Thank you for your suggestion. NRAS mutations are known to be prevalent in certain hematologic malignancies, including AML and acute lymphoblastic leukemia. Several studies have suggested that specific genetic mutations, including those involving RAS pathway genes, such as NRAS, could influence the extramedullary involvement of leukemia, potentially including the skin. The RAS pathway is involved in cell signaling, and mutations in NRAS can lead to aberrant activation of this pathway, contributing to the pathogenesis of certain cancers. We have added this point in the discussion part. 

Manuscript file (Discussion section: pages 17, lines 280-285).

Reviewer #4

In this work Owattanapanich and cols have analyzed the biological and clinical characteristics of a cohort of AML patients according to the existence of leukemia cutis (LC). To do so, they have performed a case control study obtaining 53 LC-AML patients and 106 non-LC patients. From the clinical point of view, their main findings are the significant association of LC with the presence of adenopathy, splenomegaly, hepatomegaly, low blast peripheral count and monocytic subtype. All of these factors predict the existence of LC. On the other hand, the authors found lower survival and response rate in these subgroup of extramedullary AML.

Major comments

Comment 1. What is the main endpoint of this study?

In line 61 the authors state that primary objective is to identify predictive factors discriminating patients with AML with LC from those who did not have LC. However, in the statistical analysis section (line 100) the authors say that the primary endpoint is overall survival in both AML groups. In the case OS is the main endpoint of the study, please change primary objective to in order to identify in lines 76-77. Otherwise, clarify that OS is not the main endpoint of the study.

RESPONSE: Thank you for your comment. We apologize for this mistake and have revised the manuscript to clarity. OS is the primary objective. The primary objective was to differentiate the OS outcome of patients with AML who had LC from those who did not present with LC. The secondary objective was to identify predictive factors that could distinguish between these two groups of patients. 

Manuscript file (Materials and Methods section: pages 5, lines 70-83). 

Comment 2. If one of the endpoints of this work was to analyze the predictive value of LC on overall survival, then the method for matching cases and controls is not correct, because it must be adjusted by genetics (which is the main determinant of outcome in AML). Moreover, since NGS was not done in 83% of the LC patients, there could be a bias in such a way that more LC patients harbored worse genetics and this may be the principal reason for their worse outcome. The authors recognize this limitation, but a different analysis could be made to solve this problem. For example, a propensity score matching adjusting for age, sex and genetics according ELN 2016 (or even cytogenetics alone) and performed only in the subgroup of patients treated with intensive chemotherapy could get over this limitation.

RESPONSE: Thank you for your suggestion. We agree with you. We have performed the suggested propensity score-based analysis in only the intensive chemotherapy group. A propensity score-adjusted Kaplan‒Meier survival curve by inverse probability of treatment weighted Kaplan‒Meier estimation has been performed for overall survival (OS) and relapse-free survival (RFS), comparing the outcomes between LC and non-LC groups in patient receiving intensive chemotherapy only (Supplementary figures S1 and S2). Age, sex and cytogenetic risk by the 2017 ELN classification (Döhner, et al. 2017) were fitted in a logistic regression model to develop the propensity score. The confounder-adjusted curves using the adjustedcurves package in R (Denz R, Klaaßen-Mielke R, Timmesfeld N, 2023). We also report effect size estimates by hazard ratio by propensity score case-control matched survival analysis adjusting for age, sex, and cytogenetic risk by 2017 ELN classification. The balance diagnostics for the propensity score case-control matched analyses are reported in a new table, Supplementary table S1. Matching was performed using the R package matchlt and the hazard ratios after propensity score matching were performed using the survival package in R. Text has been added to the Methods and Materials section in the Statistical Analysis subsection and to the Results section. We have added a sentence to the Discussion section. We thank you again for your insights that have improved our analyses.

Manuscript file (Materials and Methods section: pages 6-7, lines 106-119 and Materials and Methods section: page 7, lines 120-124. Results section: pages 15, lines 237-251. Discussion section: page 17, lines 290-291. Supplementary file: Supplementary table S1, supplementary figures S1 and S2)

Reference 

Döhner H, Estey E, Grimwade D, Amadori S, Appelbaum FR, Büchner T, Dombret H, Ebert BL, Fenaux P, Larson RA, Levine RL, Lo-Coco F, Naoe T, Niederwieser D, Ossenkoppele GJ, Sanz M, Sierra J, Tallman MS, Tien HF, Wei AH, Löwenberg B, Bloomfield CD. Diagnosis and management of AML in adults: 2017 ELN recommendations from an international expert panel. Blood. 2017 Jan 26;129(4):424-447. doi: 10.1182/blood-2016-08-733196. Epub 2016 Nov 28. PMID: 27895058; PMCID: PMC5291965.

Denz R, Klaaßen-Mielke R, Timmesfeld N. A comparison of different methods to adjust survival curves for confounders. Statistics in Medicine. 2023; 42(10): 1461–1479. doi: 10.1002/sim.9681

Comment 3. If NGS was performed in 9/53 LC patients, then the ELN2022 cannot be used for categorizing the genetic risk. Instead, the authors should categorize patients according ELN2016 or simply use cytogenetics.

RESPONSE: Thank you. I agree with your suggestion. We have changed to classify risk according to the 2017 ELN classification. There is no difference in the proportion of each risk group between LC and non-LC patients.

Manuscript file (Result: page 7, lines 132-134 and Table 3). 

Minor comment 

Comment 4. The conclusion in the abstract: “AML patients who developed LC tended to experience notably poorer prognoses” is not sustained by the data presented in the results section of the abstract. The authors should show in the abstract the results corresponding to this conclusion.

RESPONSE: Thank you so much for the good suggestion. We apologize for not present statistics that support the conclusion in the Abstract. We have added median overall survival times and median relapse-free survival times comparing leukemia cutis and non-leukemia cutis groups along with their log rank tests. The log rank test tests the null hypothesis whether there is a difference in Kaplan-Meier curves of survival probability for OS or RFS at any time point along the Kaplan-Meier curves compared (Bland & Altman, 2004). These p-values have also been added to the Results section.

Reference

Bland JM, Altman DG. The logrank test. BMJ. 2004 May 1;328(7447):1073. doi: 10.1136/bmj.328.7447.1073. PMID: 15117797; PMCID: PMC403858.

Manuscript file (Abstract section: pages 2, lines 35-38 and Results section: page 14 lines 221 and 223). 

Comment 5. Line 52 “These cutaneous manifestations are often associated with advanced disease”. The authors might specify the meaning of “advance disease. Is it poor overall survival?, high relapse rate?, poor cytogenetics?.

RESPONSE: Thank you for your comment to help us clarify this point. We have changed the related sentence to “These cutaneous manifestations are often associated with poor overall survival (OS) outcomes”.

Manuscript file (Introduction section: pages 4, lines 56-57). 

Comment 6. Line 76: Cases and controls were matched on baseline age, sex, and AML type. Please, specify what do you mean by type of AML (further in the text it is obvious that it means de novo vs secondary), but the authors should clarify this criteria the first time this concept is presented in the manuscript.

RESPONSE: Cases and controls were matched on baseline age, sex, the 2017 European LeukemiaNet (ELN) classification and AML type which is de novo or secondary. 

Manuscript file (Material and Methods section: pages 5, lines 78-80). 

Comment 7. Line 120 “Most patients in the LC group were classified as ECOG 1 (64.2%)” is already written in the previous line. Please remove the repeated sentence.

RESPONSE: Thank you so much. We have removed the duplicated text.

Comment 8. Please show a separate analysis of treatment response in patients treated with intensive chemotherapy and those treated with hypomethylating agents or low dose cytarabine.

RESPONSE: We perform separate analysis of treatment response in patients treated with intensive chemotherapy and those treated with hypomethylating agents or low dose cytarabine showed in Table 3. There were missing data on defining complete remission exclusively in the non-LC group constituting 8.8% missingness in the intensive chemotherapy treatment group. We informally conducted a best case/worst case sensitivity analysis by coding all missing values as “CR” for best case and all missing values as “not CR” for the worst case, generating p=0.654 and p=0.369, respectively (data not shown). There was 41.2% missingness exclusively in the non-LC group for the analysis of the hypomethylating agent/low-dose cytarabine treatment group. The same best case/worst case sensitivity analysis coding obtained p=0.169 and p=0.538, respectively (data not shown). Logistic regression results pooling these two groups and analyzing them separately are included in the Results section text to present the same results as effect sizes as odds ratios.

Manuscript file (Table 3 and Results section: pages 13-14, lines 207-213). 

Comment 9. PLOS authors have the option to publish the peer review history of their article (what does this mean?). If published, this will include your full peer review and any attached files. RESPONSE: We apologize for our lack of clear expression. I agree to including full peer reviewers’ comments and any attached files after published research.

---

## [Decision Letter · Decision Letter 1]

15 Jan 2024

Organomegalies as a Predictive Indicator of Leukemia Cutis in Patients With Acute Myeloid Leukemia

PONE-D-23-37425R1

Dear Dr. Owattanapanich,

We’re pleased to inform you that your manuscript has been judged scientifically suitable for publication and will be formally accepted for publication once it meets all outstanding technical requirements.

Kind regards,

Mehmet Baysal

Academic Editor

PLOS ONE

Additional Editor Comments (optional):

Dear Weerapat Owattanapanich;

We are pleased to inform you that our reviewers has been accepted and recommendedyour manuscript entitled "Organomegalies as a Predictive Indicator of Leukemia Cutis in Patients With Acute Myeloid Leukemia" has been accepted for publication.

Reviewers' comments:

Reviewer's Responses to Questions

**Comments to the Author**

1. If the authors have adequately addressed your comments raised in a previous round of review and you feel that this manuscript is now acceptable for publication, you may indicate that here to bypass the “Comments to the Author” section, enter your conflict of interest statement in the “Confidential to Editor” section, and submit your "Accept" recommendation.

Reviewer #1: All comments have been addressed

Reviewer #2: All comments have been addressed

2. Is the manuscript technically sound, and do the data support the conclusions?

Reviewer #1: Yes

Reviewer #2: Yes

3. Has the statistical analysis been performed appropriately and rigorously? 

Reviewer #1: Yes

Reviewer #2: Yes

4. Have the authors made all data underlying the findings in their manuscript fully available?

Reviewer #1: Yes

Reviewer #2: Yes

5. Is the manuscript presented in an intelligible fashion and written in standard English?

Reviewer #1: Yes

Reviewer #2: Yes

6. Review Comments to the Author

Reviewer #1: Well organized. No further comments. All critics are well responded.

Thank you.

Reviewer #2: (No Response)

7. PLOS authors have the option to publish the peer review history of their article (what does this mean?). If published, this will include your full peer review and any attached files.

Reviewer #1: No

Reviewer #2: **Yes: **Jan Vydra

---

## [Editor Report · Acceptance letter]

8 Feb 2024

PONE-D-23-37425R1 

PLOS ONE

Dear Dr. Owattanapanich, 

I'm pleased to inform you that your manuscript has been deemed suitable for publication in PLOS ONE. Congratulations! Your manuscript is now being handed over to our production team.

Kind regards, 

on behalf of

Dr. Mehmet Baysal 

Academic Editor

PLOS ONE